# Community Advantage and Individual Self-Efficacy Promote Disaster Preparedness: A Multilevel Model among Persons with Disabilities

**DOI:** 10.3390/ijerph16152779

**Published:** 2019-08-03

**Authors:** Rachel M. Adams, David P. Eisenman, Deborah Glik

**Affiliations:** 1Natural Hazards Center, University of Colorado Boulder, 1440 15th St, Boulder, CO 80309, USA; 2UCLA Center for Public Health and Disasters, Fielding School of Public Health, University of California Los Angeles, 650 Charles E Young Drive S, Los Angeles, CA 90095, USA; 3Division of General Internal Medicine and Health Services Research, David Geffen School of Medicine, University of California Los Angeles, 100 Glendon Ave, Suite 850, Los Angeles, CA 90024, USA

**Keywords:** multilevel modelling, disability, disaster, disaster preparedness, community resilience, community capital, self-efficacy

## Abstract

Disaster preparedness initiatives are increasingly focused on building community resilience. Preparedness research has correspondingly shifted its attention to community-level attributes that can support a community’s capacity to respond to and recover from disasters. While research at the community level is integral to building resilience, it may not address the specific barriers and motivators to getting individuals prepared. In particular, people with disabilities are vulnerable to disasters, yet research suggests that they are less likely to engage in preparedness behaviors. Limited research has examined what factors influence their ability to prepare, with no studies examining both the individual and community characteristics that impact these behaviors. Multilevel modeling thus offers a novel contribution that can assess both levels of influence. Using Los Angeles County community survey data from the Public Health Response to Emergent Threats Survey and the Healthy Places Index, we examined how social cognitive and community factors influence the relationship between disability and preparedness. Results from hierarchical linear regression models found that participants with poor health and who possessed activity limitations engaged in fewer preparedness behaviors. Self-efficacy significantly mediated the relationship between self-rated health and disaster preparedness. Living in a community with greater advantages, particularly with more advantaged social and housing attributes, reduced the negative association between poor self-rated health and preparedness. This study highlights the importance of both individual and community factors in influencing people with disabilities to prepare. Policy and programming should therefore be two-fold, both targeting self-efficacy as a proximal influence on preparedness behaviors and also addressing upstream factors related to community advantage that can create opportunities to support behavioral change while bolstering overall community resilience.

## 1. Introduction

Natural and human-initiated disasters are escalating in scale and frequency [1,2,3]. Climate change, coupled with expanding development in vulnerable settings, contributes to the degradation of existing natural resources that protect against hazards [4,5,6]. Additionally, a growing number of people reside in at-risk areas, such as coastlines and flood-prone river basins [6]. Global urbanization and greater population densities also increase the number of people exposed to potential hazards [7], including human-initiated threats such as acts of terrorism or disasters linked to technology [8]. Together, these factors intensify the threat of disasters, leading to greater population morbidity and mortality. 

Disasters do not impact people equally, with individuals living with disabilities disproportionately affected. Studies have demonstrated that different types of functional and health limitations pose challenges to following evacuation orders [9,10,11], as well as accessing shelters and temporary housing following a disaster [9,12]. Disasters can also exacerbate existing health problems due to a lack of food and clean water, temperature extremes, and physical and mental stress [13]. A lack of access to routine healthcare is a leading cause of mortality following a disaster [14], with research from past events demonstrating that people with physical and mental disabilities experience elevated disaster-related mortality [15]. Despite being at a greater risk of harm, research suggests that people with disabilities are less prepared for disasters than the general population [13,16,17,18,19,20]. Several studies demonstrate that people who have different types of functional and health limitations are less likely to possess household disaster supplies, to have developed an evacuation plan, and to engage in emergency trainings [13,16,17,19,20].

To support a safe response and quick recovery, disaster preparedness initiatives have traditionally focused on getting individuals prepared by promoting personal or household preparedness behaviors. Communication campaigns have emphasized that each household should possess an emergency supplies kit, develop a disaster plan, and stay informed about local hazards. Accordingly, a vast literature on preparedness has focused on individual motivators of these behaviors, with studies examining how different social cognitive factors influence personal preparedness [5,21]. Among the most studied social cognitive factors are knowledge, risk perception, prior disaster experience, government trust, self-efficacy, and response efficacy. Several of these variables have also been studied among people with disabilities, elucidating what factors might influence preparedness among this vulnerable segment of the population. In particular, studies have shown that people with disabilities possess diminished perceptions of self-efficacy and response efficacy to prepare for disasters. In a 2011 household survey conducted by the Federal Emergency Management Agency (FEMA), it was determined that only 65% of Americans who self-report as having a disability believe they know how to get prepared [22]. This survey also found that in comparison to people without disabilities, those with a disability were less likely to perceive that they could respond to different types of hazards and disasters [22]. When asked whether they believe preparing for disasters helps, people with disabilities were less likely to believe in the effectiveness of preparedness behaviors across a wide range of emergencies, including natural disasters, terrorist acts, and different hazards such as wildfires and floods [22]. In another study that used an open-ended questionnaire to assess barriers to preparedness among people with mobility impairments who had previously experienced a disaster, personal accounts from survivors highlighted how people with disabilities had lost confidence in their ability to keep themselves safe [23]. These findings suggest that disabilities may influence perceptions about effectively engaging in behaviors that can protect against hazards, which may in turn influence whether or not they prepare.

More recently, disaster preparedness and planning initiatives have shifted focus away from individual behaviors to emphasize community-level resilience building. Community resilience has become the central framework guiding disaster preparedness and planning efforts, with researchers, practitioners, and policy makers increasingly shifting their attention to processes and factors that can increase community capacity to mitigate threats from diverse hazards [24]. Indeed, both governmental and nongovernmental organizations have assumed a community resilience framework to guide disaster planning programs, such as the Rockefeller Foundation’s 100 Resilient Cities program and the Centers for Disease Control and Prevention’s Public Health Emergency Preparedness Program. Disaster preparedness research has thus increasingly focused on different ways to define, measure and promote resilience at the community level. When treated as a process that links a community’s dynamic resources to adaption after a disaster, researchers have demonstrated that communities with more advanced economic development and social capital are more resilient [25,26]. When focusing on ways to engage community stakeholders in activities that can help build resilience, studies have found that buy-ins from local leadership, shared decision-making among citizens, civic engagement, and neighborhood social capital enhance participation in community planning behaviors, such as attending community-based meetings and trainings [27,28]. 

To reduce preparedness discrepancies among people living with disabilities, we need to identify what factors influence their participation in these behaviors. It is important to consider lessons learned from past research by focusing on individual social cognitive pathways that explain why different types of disabilities interfere with preparedness behaviors. Additionally, we need to understand what community-level factors enhance preparedness, in order to be able to continue to invest in community resilience initiatives that address barriers faced by people with disabilities, while strengthening a whole community’s capacity to respond to and recover from disasters. Multilevel modeling offers a unique perspective that can be used to examine how both individual- and community-level factors impact preparedness behaviors. 

Using hierarchical linear modeling, we assessed social cognitive pathways and community characteristics to explain why people with disabilities are less prepared for disasters. We analyzed individual-level data from the 2013 Public Health Response to Emergent Threats Survey—a household survey conducted in 16 different communities in Los Angeles County, linked with community-level data focusing on measures of community advantage from the Healthy Places Index. Los Angeles County, the study setting, is threatened by a number of natural disasters, including earthquakes, wildfires, mudslides, and tsunamis, as well as human-initiated threats such as technological hazards and acts of terrorism. It is also a region with large social and economic disparities, making it a valuable setting to study how community-level attributes influence individual disaster preparedness behaviors among vulnerable members of the population. Results from this study can be used to guide more inclusive and targeted preparedness strategies and ultimately promote effective disaster response and recovery among people living with disabilities.

## 2. Hypotheses

We propose the following study hypotheses:

H1. There is an inverse relationship between disability status and disaster preparedness such that people living with disabilities participate in fewer disaster preparedness behaviors in comparison to those who do not have disabilities.

H2. Self-efficacy for preparing for disasters partially mediates the relationship between living with disabilities and disaster preparedness, with one’s disability status associated with lower self-efficacy, a positive correlate of preparedness.

H3. Response efficacy for preparing for disasters partially mediates the relationship between living with disabilities and disaster preparedness, with disability status associated with lower response efficacy, a positive correlate of preparedness.

H4. Community advantage moderates the association between disability status and disaster preparedness such that when advantage is high, the negative relationship between disabilities and preparedness is weaker; conversely, when advantage is low, the negative relationship between disabilities and preparedness is stronger. 

These hypotheses are summarized in the conceptual model depicted in Figure 1.

## 3. Materials and Methods

### 3.1. The Public Health Response to Emergent Threats Survey

The Public Health Response to Emergent Threats Survey (PHRETS) is periodically conducted in Los Angeles County to guide and evaluate local disaster planning efforts. The 2013 survey was specifically designed to collect baseline data as part of the mixed methods evaluation strategy for the Los Angeles Community Disaster Resilience Project (LACCDR). The LACCDR was a community-based program that involved the implementation of a community resilience toolkit equipped with training resources and technical assistance to specific communities. However, because we used baseline data prior to the implementation of any experimental conditions, we treated the data as a cross-sectional assessment of the population within the communities included in the program. PHRETS survey questions were drawn from published studies and planned national surveys, including the Behavioral Risk Factor Surveillance System and the FEMA’s 2009 Citizen Corps Survey.

A sample of 16 communities in Los Angeles County was chosen based on criteria for defining cohesive communities, including having a local business community, a school/school district, police and fire department services, a community clinic/hospital/health responsible entity, or evidence of engaged community-based organizations or civic leaders. We used address-based sampling as a frame to identify residential mail delivery locations in the communities. The sampling unit was the address and we randomly selected households using computer-generated random sampling. Adult residents (≥18) were invited to participate in the survey, with a total of 4700 respondents completing interviews (response rate = 35%).

PHRETS was conducted via telephone in English, Spanish, and Korean between 3 June and 7 August, 2013. Trained interviewers utilized a computer-assisted telephone interviewing system. The study protocol was reviewed and approved by the institutional review board of the Los Angeles County Department of Public Health prior to data collection. 

### 3.2. The Healthy Places Index

The Health Paces Index (HPI) is an index measuring community advantage [29]. It contains 25 indicators related to the social determinants of health that can be organized into the following eight policy action domains: economy, education, healthcare access, housing, social environment, neighborhood conditions, clean environment, and transportation [29]. Percentile ranks are publicly available for each indicator, domain, as well as a composite HPI measure. Higher percentile rankings indicate a greater level of advantage. HPI includes measures at the census tract level for all California census tracts where there is a population of 1500 or greater in the 2010 decennial census and where the group quarters’ population is less than 50% of the total 2010 population [29].

This study uses the HPI composite percentile ranks and domain percentile ranks for the census tracts in the Los Angeles communities included in the PHRETS sample. The composite percentile rank adds the domain ranks in weighted fashion, with domain weights estimated via the weighted quantile score method using life expectancy at birth as the independent outcome [29]. Indicators included in the HPI come from different publicly available data sources published by government institutions, non-profits, and universities. More information about the development of the HPI is described on its website (https://healthyplacesindex.org/). 

### 3.3. Measures

#### 3.3.1. Disability

The World Health Organization’s International Classification of Functioning, Disability and Health (ICF) provides a common language and framework for describing disability and functional status. Unlike traditional health indicators of morbidity and mortality that have been seen as distinct from disability, the ICF conceptualizes human functioning as a synthesis of biological, psychological, social, and environmental aspects of health [30]. Disability is broadly conceptualized as the interaction between an individual’s health condition and their personal and environmental setting, and includes any physical or mental impairment that limits participation in activities or life situations [31]. Guided by the ICF, we characterize disability as a term that encompasses impairments, activity limitations, and restrictions in the involvement of life situations [31]. People living with disabilities are thus members of the population who experience health conditions that interfere with functioning. This broad but meaningful conceptualization includes people who experience physical, mental or emotional limitations and is meant to be used for decision-making purposes [31], including applications of disaster research. 

As this study involves the operationalization of the disability construct, we chose to measure disability using four separate variables of health restrictions from the PHRETS survey. The first measure assessed self-rated health by asking respondents “In general, would you say that your health is excellent, very good, good, fair, or poor?” The following three dichotomous questions were also asked: (1) Are you limited in any way in any activities because of a physical, mental, or emotional problems?; (2) Do you now have any health problems that require you to use special equipment such as a cane, a wheelchair, a special bed, or a special telephone?; and (3) Do you consider yourself a person with a disability?

#### 3.3.2. Disaster Preparedness

The PHRETS collected data on both personal and community preparedness behaviors. To assess personal preparedness behaviors, the survey asked respondents whether they had a 3-day supply of water, a 3-day supply of non-perishable food, a household reunification plan, and whether in the past 12 months they looked for information about getting prepared for a disaster or bought additional emergency supplies. To assess community preparedness, the survey asked whether in the past 12 months they attended any training to help others in the community in a disaster, attended a community meeting where preparing for disasters was discussed, worked or volunteered to help the community prepare for or respond to a disaster, or attended any training in Psychological First Aid. Assuming a holistic approach to preparedness that both promotes self-sufficiency and community engagement in disaster planning, we summed together these 10 different types of behaviors into a single disaster preparedness index. Together, these different types of behaviors promote an entire community’s capacity to respond to and recover from a disaster, contributing to its ongoing development of community resilience [32]. 

#### 3.3.3. Self-Efficacy and Response Efficacy

Self-efficacy, or confidence about one’s ability to effectively engage in a behavior, is an important social cognitive precursor to preparing for disasters. In Paton’s social-cognitive preparation model [33], which adapts elements of Bandura’s social cognitive theory [34] to the context of disasters, self-efficacy acts as one of the key constructs influencing intentions to prepare. He posits that people will develop intentions to prepare only if they have adequate expectations about being able to perform the behavior. Response efficacy denotes perceptions about how engaging in protective behaviors will effectively lead to intended outcomes. Often additionally referred to as outcome efficacy or behavioral efficacy, it depicts beliefs about the effectiveness of behaviors to promote health or protect against risk. As described in Paton’s social-cognitive preparation model, response efficacy acts as a precursor to actually engaging in disaster preparedness behaviors [33].

Two variables from the PHRETS dataset were used to measure self-efficacy and one variable for response efficacy. Each of these measures used a 4–point Likert scale ranging from strongly disagree to strongly agree. The self-efficacy questions asked respondents about their level of agreement with the following statements: (1) I am confident I can protect and help myself in the event of a disaster, such as an earthquake; and (2) I am confident I can be of help to my neighbors or community in the event of a disaster, such as an earthquake. Together these different measures were intended to assess one’s confidence in one’s ability to effectively engage in both personal and community preparedness behaviors. Responses were combined with equal weights into a single continuous index (Cronbach alpha = 0.751). Response efficacy was measured by asking respondents about their level of agreement with “I don’t think it really matters if you plan for disasters, such as a major earthquake”. 

#### 3.3.4. Community Advantage

While there are many different definitions of community resilience throughout the hazards and disaster literature, there is a general consensus among researchers and practitioners that one of its defining characteristics is its multidimensionality [28]. Different conceptualizations of community resilience emphasize that communities have different types of resources or capitals that they can leverage to adapt to external shocks and stressors. The dimensions of community resilience typically include environmental conditions, the built infrastructure, economic assets, human and cultural capital, social capital, and political power [28]. Communities with more advantaged conditions across these different domains are considered more resilient to disasters. 

Guided by a community resilience perspective, we used the Healthy Places Index to measure community advantage. While not specifically developed within the context of disasters, the HPI summarizes the conditions and levels of key community resources that foster a healthy population and health equity [29]. The HPI is comprised of 25 individual indicators that can be organized and ranked according to 8 domains of community advantage (economy, education, healthcare access, housing, neighborhoods, clean environment, transportation, and social environment). Greater advantage across these domains indicates a healthier community that can thrive in the face of adversity. 

We operationalized community advantage using the HPI domain scores as well as a composite HPI score aggregated to the community level. The economy domain consists of measures of median household income, the percent of the population greater than 200% the Federal Poverty Level, and the percent of the population aged 25–64 years who are employed. The education domain consists of the percent of people 25 or older who have at least a bachelor’s degree, the percent of the population aged 15–17 years that are enrolled in high school, and the percent of 3–4 year-olds who are enrolled in pre-school. The healthcare access domain is comprised of one variable measuring the percent of the population aged 18–64 years which is insured. The housing domain has measures of the percent of property owners, percent of households with full kitchen facilities and plumbing, percent of low-income homeowners paying more than 50% on housing, percent of low-income renters paying more than 50% on housing, and households with one occupant or less per room. The social domain contains the percent of registered voters and the percent of households with children where two parents are present. Neighborhood condition is comprised of the percent of the population living within 0.5 miles of green space, the percent of the census tract tree canopy, the percent of urban (or rural) population living with 0.5 (or 10) miles of a supermarket, the percent of the population living within 0.25 miles of an alcohol outlet, and the employment density for retail, entertainment or education. The clean environment domain has measures of gridded diesel PM emissions, the water contaminant index, the mean 8-hour ozone concentration in summer, and the annual mean concentration of PM_2.5_. Finally, the transportation domain is made up of the percent of households with access to an automobile and the percent of workers commuting by walking, cycling, or transit. 

#### 3.3.5. Demographic Characteristics

We additionally analyzed age (18–29, 30–44, 45–59, 60+), gender (male or female), and race/ethnicity (white, African American, Asian, Hispanic, other) variables that were collected by the PHRETS.

### 3.4. Data Analysis

Prior to analysis, the HPI data were aggregated to the community level so that it could be merged with the PHRETS dataset. We used the U.S. Department of Housing and Urban Development USPS zip code crosswalk files to match the PHRETS sample zip codes to corresponding census tracts. We used a ratio of residential addresses in the census tract to all the residential addresses in the zip code, which allowed us to weight the census tract HPI scores in each zip code. The weighted census tract values were summed to derive zip code measures, which were used to calculate average HPI scores for each community.

To test the association between each of the four disability measures and the disaster preparedness outcome, hierarchical linear regression was conducted with individual respondents nested within communities. We only included an error term for the intercept at level 2, as it was assumed that the slope was non-randomly varying at level 2 and including a slope error term at level 2 was non-significant. Individual age, race/ethnicity and gender were included as level 1 covariates in each of the models. Income and education were not included as covariates due to a high correlation with the community. Multicollinearity between covariates was assessed by generating Variance Inflation Factor statistics between different covariate pairs. To assess model fit, we conducted deviance tests between nested models and generated intraclass correlation coefficients. 

To evaluate whether self-efficacy or response efficacy mediated the relationship between disability and disaster preparedness, we conducted mediation analysis in hierarchical linear regression models based on Baron and Kenny’s product method (Figure 2). We conducted this analysis separately for each disability measure and the two proposed mediators—self-efficacy and response efficacy. In Figure 2, X is the independent variable (disability), M is the mediating variable (self-efficacy or response efficacy), and Y is the outcome variable (disaster preparedness). The first step was to establish a statistically significant relationship between the focal independent and outcome variable (path c). We then determined whether there was a relationship between the independent variable and the mediation variable (path a) by regressing self-efficacy or response efficacy on each of the disability variables. Next, we established whether there was a significant positive association between the mediator variable and the outcome variable (path b). This model contained both the independent variable and the mediator. Finally, using this same regression model, we assessed the net direct effect of the independent variable on the outcome variable while taking into account the indirect effect of the mediator (c’). We hypothesized that both self-efficacy and response efficacy contribute to partial mediation, so the direct effect of the focal relationship coefficient (c) will reduce in magnitude with the addition of the mediator (i.e., c’< c) but still remain statistically significant. The significance of the mediation was assessed via the Sobel Test. We hypothesized that the different disability measures would be negatively associated with self-efficacy and response efficacy, and that these variables would be positively associated with the disaster preparedness outcome. We therefore assumed that these mediators would partially mediate the relationship between disability and preparedness. 

To test whether community advantage moderated the relationship between disability and preparedness, we ran separate hierarchal linear regression models that included an interaction term with the disability measures. We hypothesized that community advantage would attenuate the negative relationship between disability and preparedness by having a significant and positive interaction coefficient (α = 0.05). We conducted moderation analyses using both the composite HPI score and each of the HPI domains. The following equations represent the model used to test this hypothesis: 

Level 1: Y_ij_ = b_0j_ + b_1j_X_ij_ + e_ij_

Level 2: b_0j =_ g_00_ + g_01_HPI_j_ + u_0j_

b_1j_ = g_10_ + g_11_HPI_j_

where Y_ij_ = disaster preparedness outcome for the ith individual in the jth community 

b_0j_ = level 1 intercept

b_1j_ = level 1 slope for the ith individual in the jth community

X_ij_ = disability status for the ith individual in the jth community 

e_ij_ = within community error term

HPI_j_ = community-level advantage measures via HPI percentile

g_00_ = average intercept across communities 

g_01_ = level 2 slope

u_0j_ = level 2 intercept error term

g_10_ = average slope across communities

g_11_ = level 2 slope of interaction term. 

When combined into a single formula, the interaction term becomes apparent:

Y_ij_ = g_00_ + g_01_HPI_j_ + g_10_ X_ij_ + g_11_ X_ij_*HPI_j_ + (u_0j_ + e_ij_).

Missing data were treated with listwise deletion, as none of the covariates had missing values greater than 5%. Each of the models contained individual and community-level weights. The individual weights were estimated using raking ratio estimation with population benchmarks for age, sex, and race/ethnicity based on the 2010 census blocks for each of the 16 communities. To reduce bias in multilevel analysis, the level 1 weights were scaled to the sum of the actual number of completed cases in each level 2 cluster [35]. All analyses were conducted using SAS version 9.4 (SAS Institute Inc.: Cary, NC, USA). 

## 4. Results

Table 1 presents the socio-demographic characteristics of the study population after individual weighting. The majority of respondents were female (52%), between 30 and 44 years old (31%), white (40%), possessed at least a college degree (34%), earned a household income between $10,000 and $30,000, and spoke mostly English in the home (57%). Generally, these demographics reflect similar distributions of gender, age, and race/ethnicity in Los Angeles County based on data from the 2010 Census [36]. However, the sample tended to have a lower proportion of Hispanic respondents, and more people that possessed higher education (both high school and college graduates), spoke English in the home, and had a lower household income than the general Los Angeles County population [37].

Results from the self-efficacy and response efficacy mediation analyses are located in Table 2. As demonstrated from step 1, which assessed whether there is a statistically significant association between each of the disability measures and disaster preparedness, only worse self-rated health (B=-0.23, *p* < 0.01) and activity limitations (B = −0.23, *p* = 0.03) were negatively associated with preparing for disasters. Requiring the use of special medical equipment and considering yourself a person with a disability were not significantly associated with preparedness. 

For self-rated health, step 2 demonstrated that worse health was associated with lower self-efficacy (B= −0.25, *p* < 0.01) and response efficacy (B = −0.06, *p* < 0.01). Both of these variables were also significantly positively associated with preparedness and including them in the model reduced the magnitude of the self-efficacy coefficient. Sobel tests later confirmed that all else being equal, worse self-rated health was associated with a mean decrease of 0.15 disaster preparedness behaviors, as mediated by self-efficacy. The mediation was non-significant for response efficacy.

The activity limitations variable was negatively associated with self-efficacy. When we added self-efficacy to the preparedness model, the activity limitations coefficient also reduced in magnitude from B = −0.23 to B = −0.06 and became non-significant (*p* = 0.54). This suggests that self-efficacy contributes to full mediation; however, results from the Sobel test found that mediation was not statistically significant. The presence of activity limitations was not significantly associated with response efficacy so no mediation was occurring.

Because the variables measuring the presence of a health problem requiring the use of special medical equipment and considering yourself a person with a disability were not significantly associated with disaster preparedness, they were dropped from further analysis. The results from the community advantage moderation analyses with self-rated health (Model 1) and activity limitations (Model 2) are located in Table 3. In Model 1, which examined whether community advantage moderated the relationship between self-rated health and preparedness, the interaction term with poor self-rated health was statistically significant. All else equal, for every one unit increase in community advantage, individuals with poor self-rated health participated in a mean increase of 0.02 disaster preparedness behaviors. 

In Model 2, which examined whether community advantage moderated the relationship between activity limitations and preparedness, the interaction term was non-significant. Community advantage therefore did not significantly moderate the relationship between activity limitations and preparedness. 

Results from the moderation analyses using the HPI domains and self-rated health are located in Table 4. The effect of poor health on preparedness was significantly attenuated by living in a community with more advantaged housing and social attributes. For every one unit increase in the housing domain, individuals with poor self-rated health participated in a mean increase of 0.02 disaster preparedness behaviors. For every one unit increase in the social domain, individuals with poor self-rated health participated in a mean increase of 0.02 disaster preparedness behaviors. 

Deviation analyses were conducted for each of the multilevel models and smaller, nested models without a level 2 intercept to establish whether a multilevel model possessed a better fit than a single level model. In each deviation analysis, the chi-square value generated from a likelihood ratio test was statistically significant (*p* < 0.05), justifying the use of multilevel analysis. The intraclass correlation coefficients for each of the models varied between 0.07 and 0.09. These values were very low, signifying that the vast majority of the variation in preparedness behavior was within communities and not between communities.

## 5. Discussion

Using a multilevel approach, this study demonstrates that both individual- and community-level factors influence preparedness among a vulnerable segment of the population. First, we confirmed findings from past research that more general perceptions of health and activity limitations are associated with participating in fewer preparedness behaviors [13,16,17,18,19,20]. Using mediation analysis, we revealed that lower perceived self-efficacy partially explains this discrepancy. People with worse self-rated health have lower confidence in their ability to engage in protective behaviors, which partially contributes to lower levels of preparedness. Furthermore, community-level advantage has a positive impact on behavior. Individuals with poor self-rated health who live in more advantaged communities prepare more for disasters than those living in less advantaged communities. The housing and social environments appear to be driving this moderation effect. A revised conceptual model summarizing these finding is presented in Figure 3.

The results provide evidence supporting the hypothesis that people with disabilities are less prepared, but for only two of the four measures of disability. While worse self-rated health and the presence of activity limitations were significantly negatively associated with preparing for a disaster, there was no significant relationship for possessing a health problem that requires the use of special medical equipment or considering yourself to be a person with a disability. These mixed results suggest that more general perceptions of health restrictions may be a better indicator of one’s ability to prepare. For instance, self-rated health is a subjective assessment of health and wellbeing that not only captures a broad range of conditions but has also been linked to other factors related to disaster vulnerability, such as a poor quality of interpersonal relationships and lower income and education [38]. It is possible that people with worse general health are overburdened with their conditions and may experience fatigue, reduced confidence in their abilities, or limited resources to gather supplies, develop plans, or work with members of their community to prepare [13]. The presence of activity limitations also acts as a proxy measure for having health conditions or disabilities that interfere with regular functioning. The presence of conditions that limit participation in daily activities might also hinder involvement in preparedness behaviors, particularly for activities requiring more time and effort such as attending community planning events. 

The null findings for both having a health problem requiring the use of special medical equipment and considering yourself to be a person with a disability may result from the low frequency of these measures in the sample. The PHRETS collected data from the general population and the low prevalence of these more extreme measures may have not provided enough variance in the outcome to detect significant effects. Additionally, the way the questions were phrased suggests potential weaknesses in these measures. For the medical equipment question, respondents were not asked about other types of medical dependence beyond the specific types of equipment listed in the prompt (e.g., cane, wheelchair, special bed, or a special telephone) and thus may have neglected to capture individuals with more common chronic conditions that require equipment (e.g., inhalers or insulin syringes). To consider oneself to be a person with a disability also requires that an individual’s perceptions of disability coincide with existing functional and health limitations. It is possible that people living with different types of disabilities do not consider themselves to be disabled, as they do not possess physical impairments that are typically associated with making a social security disability claim. The potential weaknesses in these variables coupled by their low frequency in the sample population may have therefore contributed to their non-significant relationship with preparing for disasters.

Self-efficacy, or perceptions about one’s ability to effectively engage in a behavior, is a well-studied correlate of disaster preparedness. While less studied, research also suggests that people with disabilities are less likely to perceive that they could respond to different types of hazards and disasters [22]. Findings from this study support past literature by demonstrating that self-efficacy is positively associated with disaster preparedness and negatively associated with each of the four measures of disability. Results from the mediation analysis additionally confirm that self-efficacy partially mediates the relationship between self-rated health and preparing for disasters, suggesting that diminished confidence in the ability to prepare for disasters acts as a barrier among people in poor health. Conversely, response efficacy was not a significant mediator of the relationship between disability and preparedness. While believing in the effectiveness of preparedness behaviors to protect against hazards was positively associated with preparing for disasters in general, it did not explain why people with worse health or activity limitations are less prepared. More proximal perceptions about being able to engage in protective behaviors thus appear to be more influential among people with disabilities than beliefs about whether preparedness influences disaster outcomes.

The interaction analysis demonstrated that community advantage moderated the relationship between poor self-rated health and preparing for disasters. For every one unit increase in community advantage, individuals with poor self-rated health participated in a mean increase of 0.0155 disaster preparedness behaviors. This significant finding was quite surprising due to the low intraclass correlation coefficient for the self-rated health model. Less than 10% of the variation was between communities, so the fact that a community-level variable could still detect a moderation effect suggests it had a strong influence on preparedness among people in poor health. This finding supports the rationale that living in a community with greater advantage provides a number of additional opportunities and resources that support preparedness. 

Results from the sub-analyses using the HPI domains suggest that this moderating effect is mostly driven by the community’s housing and social attributes, followed by marginal effects from economic characteristics, healthcare access, and the cleanliness of the environment. The HPI’s housing domain contains measures of crowding, percent of homeowners, cost burden of housing, and higher quality housing. There are several possible reasons that could explain why the housing domain reduced the negative influence of poor health on preparedness behaviors. First, as homeowners, people may be more invested in the security of their home and thus more likely to stockpile emergency supplies. This is supported by past research that has found that compared to renters, homeowners are more likely to stock the set of recommended emergency items [39] and prepare the household for a disaster [40]. Homeownership can also denote greater stability in the composition of neighborhood residents, which may mean they are more likely to possess prior experience with local emergencies, a strong motivator to being prepared [41]. Better quality housing and a lower cost burden of housing additionally suggest better community economic conditions, which may contribute to individual financial resources to purchase and maintain disaster supplies. It may also mean greater community resources to host emergency training that residents can attend. People with poor health who live in communities with overall better housing conditions may therefore possess greater motivation and access to resources that can foster disaster preparedness. 

The HPI’s social domain is comprised of the percent of registered voters and the percent of family homes with two parents. Going beyond a community’s economic characteristics, this domain captures more of the social involvement of community members, both within the household and relating to civic responsibility. There are several plausible interpretations for why the social domain mitigated the negative association between poor health and preparedness. To start, a community where more people are registered to vote signifies greater political engagement. People who want to make a difference in their communities through political means may also be more inclined to engage in community-building activities, such as attending community disaster meetings and trainings. The presence of two parents in the home, which has historically been linked to such factors as lower rates of poverty [42,43] and greater academic achievement among children [44,45,46], can also be thought of as a proxy measure for both familial stability and support. Communities with supportive family structures may foster greater planning for disasters among members of the family. This may be particularly influential for people in poor health, who may rely more on others during emergency situations.

### Limitations

Several study limitations must be acknowledged. First, the data were cross-sectional, so we cannot make causal inferences. While it is plausible that functional and health limitations can reduce participation in disaster preparedness behaviors, all of the findings must be interpreted as association rather than causation. Second, the intraclass correlation coefficients (ICC) for the multilevel models were low, signifying that most of the variation was within communities and not between communities. Though deviance tests still justified the use of multilevel modeling, the low ICC affected the power needed to assess how community attributes influence preparedness among people with disabilities. Third, the PHRETS survey respondents were only sampled from households within the designated Los Angeles County communities, contributing to selection bias. Not only did sampling from the general population contribute to a low frequency of people with disabilities, but there was likely a systematic exclusion of people with more severe disabilities, as they are more likely to live in group quarters or assisted living environments that were not included in the sampling frame. By excluding respondents who did not live independently, the results cannot be generalizable to people with more severe intellectual and physical disabilities. Fifth, all of the preparedness variables included in the outcome measure were asked from an all-hazards perspective. This type of disaster may elicit different coping mechanisms among people with different functional and health limitations. Future research should consider assessing how different types of disability influence preparedness behaviors across specific types of natural and human-initiated disasters. Finally, the PHRETS uses self-report measures, which may have contributed to response bias, particularly when dealing with more abstract concepts. Future disability research should use more objective measures, such as medical chart data, to study the different ways that health may interfere with preparing for disasters. 

## 6. Conclusions

This study focuses on a timely and crucial public health topic that can help advance efforts to prevent disaster-related morbidity and mortality. Not only are disasters increasing in scale and frequency, but research suggests that emergency management agencies are not prepared to address the diverse needs of people with disabilities [47,48,49]. Using multilevel analysis, the study also highlights that both individual and community factors are essential to influencing disaster preparedness among people with disabilities. We therefore recommend that policy and programming be two-fold—both targeting more proximal cognitive precursors to preparedness behaviors and also addressing upstream factors related to community advantage that can create opportunities to support behavioral change while bolstering overall community resilience. 

In terms of the individual social cognitive characteristics, the results demonstrate that lower self-efficacy acts as an inhibiting factor to getting people with disabilities prepared. Preparedness initiatives should therefore target this intermediary to influence behavioral change. For instance, educational programs targeting people with disabilities should adapt theory-informed best practices for strengthening self-efficacy, such as social modeling and verbal persuasion [50]. Disaster preparedness communications, such as messages disseminated via mass communication channels or more locally in settings where people with disabilities may frequent—such as pamphlets and posters distributed at hospitals or nursing homes—should also use language that fosters perceptions of self-efficacy among recipients. With a whole literature that focuses on building self-efficacy across different public health disciplines, there are a number of opportunities to translate these findings to the field of disaster preparedness and response.

The results also emphasize the importance of community in influencing preparedness behaviors. Community advantage helped mitigate the negative influence of poor health on preparing for disasters, suggesting the need to invest more in strengthening overall communities, particularly in the areas of housing and the social environment. In recent years, both government and nongovernment agencies have started implementing community resilience programs that aim to improve a community’s ability to protect against, mitigate, respond to, and recover from diverse threats and hazards. For instance, the Rockefeller Foundation’s 100 Resilient Cities program partners with cities across the globe to support their development of different resilience-building strategies. In Los Angeles, these strategies have expanded beyond individual preparedness to focus on enhancing the built infrastructure, community connectedness, and interdisciplinary partnerships [51]. The results from this study support several of the goals that were developed by Los Angeles through the 100 Resilient Cities program, including the building of affordable housing and the strengthening of civic engagement through community collaborations [51]. Communities could strengthen housing by such actions as increasing the production and preservation of affordable housing, promoting and expanding housing for vulnerable populations, and developing neighborhood-based post-disaster housing plans [51]. Civic engagement could be promoted through neighborhood outreach and education about local risks, partnerships between community organizations and local experts and resources, and the promotion of neighborhood disaster planning programs [51]. Continued investment in communities not only provides people with disabilities with the additional resources needed to improve preparedness behaviors, but it can help build communities that are resilient to the impact of disasters.

## Figures and Tables

**Figure 1 ijerph-16-02779-f001:**
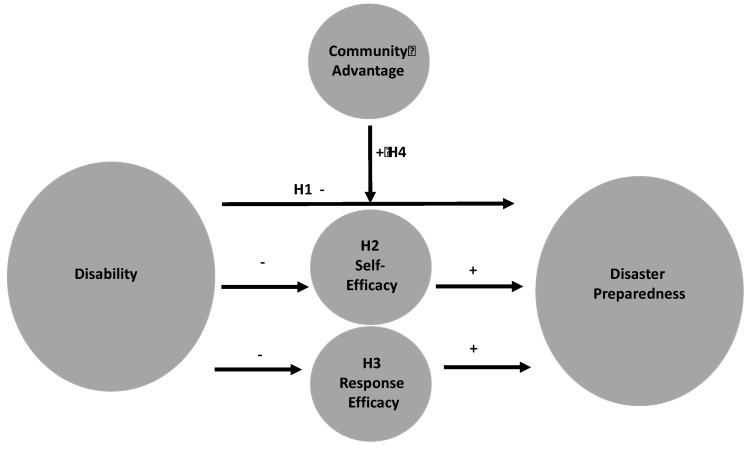
Conceptual model linking disability and disaster preparedness.

**Figure 2 ijerph-16-02779-f002:**
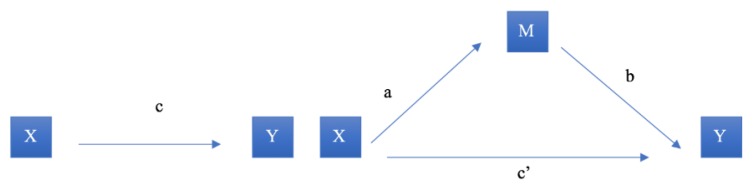
Mediation Diagram Based on Baron and Kenny’s (1986) Product Method.

**Figure 3 ijerph-16-02779-f003:**
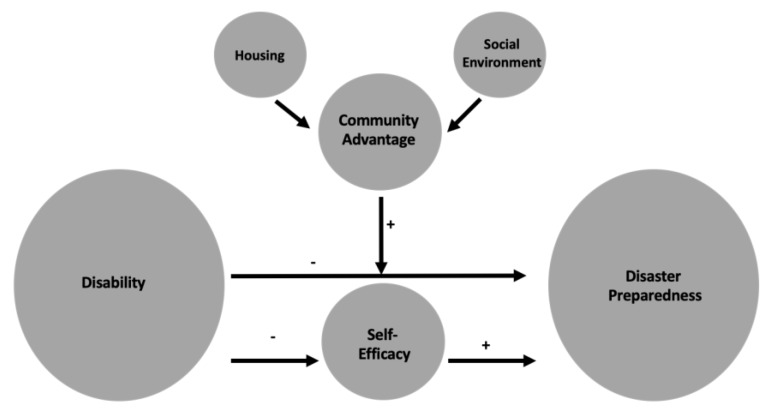
Revised conceptual model linking disability and disaster preparedness.

**Table 1 ijerph-16-02779-t001:** Sample characteristics of respondents after weighting, Public Health Response to Emergent Threats Survey, 2013 (*n* = 4700).

Variables		*n* (%)
Gender	Male	2274 (48%)
	Female	2418 (52%)
Age	18–29	1026 (22%)
	30–44	1445 (31%)
	45–59	1260 (27%)
	60+	941 (20%)
Race/ethnicity	White	1829 (40%)
	African American	440 (10%)
	Asian	471 (10%)
	Hispanic	1537 (34%)
	Other	239 (5%)
Income	<$10,000	444 (11%)
	$10,000–29,999	1157 (29%)
	$30,000–49,999	842 (21%)
	$50,000–99,999	928 (23%)
	>$100,000	652 (16%)
Education	Some high school or less	691 (15%)
	High school graduate/GED	1164 (25%)
	Associate degree/trade school/some college	1243 (26%)
	College degree or higher	1602 (34%)
Language	English	3068 (66%)
	Spanish	1262 (27%)
	Other	310 (7%)

**Table 2 ijerph-16-02779-t002:** Mediation analysis of self-efficacy and response efficacy on the relationship between disability and disaster preparedness, Los Angeles County, 2013 (*n* = 4700) ^a^.

Independent Variable	Coefficient (*p* value)				
	Self-efficacy				
	Step 1 (c)	Step 2 (a)	Step 3 (b)	Step 4 (c’)	Mediation a*b
Worse self-rated health	−0.23 (*p* < 0.01)	−0.25 (*p* < 0.01)	0.58 (*p* < 0.01)	−0.08 (*p* = 0.04)	−0.15 (*p* < 0.01)
Limited in activities from physical, mental, or emotional problems	−0.23 (*p* = 0.03)	−0.32 (*p* < 0.01)	0.59 (*p* < 0.01)	−0.06 (*p* = 0.54)	−0.19 (*p* = 1.00)
Presence of health problem requiring use of special medical equipment	−0.09 (*p* = 0.44)	−0.33 (*p* < 0.01)	0.60 (*p* < 0.01)	0.07 (*p* = 0.49)	N/A
Consider yourself a person with a disability	−0.01 (*p* = 0.94)	−0.38 (*p* < 0.01)	0.60 (*p* < 0.01)	0.21 (*p* = 0.12)	N/A
		Response Efficacy			
	Step 1 (c)	Step 2 (a)	Step 3 (b)	Step 4 (c’)	Mediation a*b
Worse self-rated health	−0.23 (*p* < 0.01)	−0.06 (*p* < 0.01)	0.21 (*p* = 0.01)	−0.22 (*p* < 0.01)	−0.01 (*p* = 0.99)
Limited in activities from physical, mental, or emotional problems	−0.23 (*p* = 0.03)	−0.01 (*p* = 0.66)	0.23 (*p* < 0.01)	−0.23 (*p* = 0.04)	N/A
Presence of health problem requiring use of special medical equipment	−0.09 (*p* = 0.44)	0.05 (*p* = 0.13)	0.24 (*p* < 0.01)	−0.07 (*p* = 0.56)	N/A
Consider yourself a person with a disability	−0.01 (*p* = 0.94)	−0.07 (*p* = 0.04)	0.23 (*p* < 0.01)	0.02 (*p* = 0.90)	N/A

^a^ Hierarchical linear modelling controlling for gender, age and race/ethnicity.

**Table 3 ijerph-16-02779-t003:** Disaster preparedness vs. disability measures with community advantage moderation, Los Angeles County, 2013 (*n* = 4700) ^a^.

Independent Variable	Coefficient	*p* Value
Model 1		
Self-rated health		
Poor	−1.32	< 0.01
Fair	−0.61	0.02
Good	−0.15	0.52
Very good	−0.05	0.75
Excellent (reference)	--	--
Community advantage	0.00	0.65
Self-rated health*community advantage		
Poor*community advantage	0.02	0.02
Fair*community advantage	−0.00	0.68
Good*community advantage	−0.01	0.14
Very good*community advantage	−0.00	0.78
Excellent*community advantage (reference)	--	--
Model 2		
Presence of activity limitations		
Yes	−0.43	0.09
No (reference)	--	--
Community advantage	0.00	0.82
Presence of activity limitations*community advantage		
Yes*community advantage	0.01	0.28
No*community advantage (reference)	--	--

^a^ Hierarchical linear modelling controlling for gender, age and race/ethnicity.

**Table 4 ijerph-16-02779-t004:** Disaster preparedness vs. self-rated health with the HPI ^a^ domain interactions, Los Angeles County, 2013 (*n* = 4700) ^b^.

HPI Interaction Term with Self-Rated Health	Coefficient	*p* Value
Poor*Economy	0.01	0.05
Fair*Economy	0.00	0.82
Good*Economy	−0.01	0.20
Very Good*Economy	0.00	0.92
Poor*Education	0.01	0.23
Fair*Education	0.00	0.54
Good*Education	0.00	0.89
Very Good*Education	0.01	0.14
Poor*Healthcare access	0.02	0.09
Fair*Healthcare access	0.00	0.72
Good*Healthcare access	0.00	0.75
Very Good*Healthcare access	0.01	0.09
Poor*Housing	0.02	< 0.01
Fair*Housing	0.00	0.81
Good*Housing	0.00	0.91
Very Good*Housing	0.01	0.09
Poor*Neighborhood	−0.01	0.68
Fair*Neighborhood	0.01	0.30
Good*Neighborhood	0.01	0.22
Very Good*Neighborhood	0.01	0.44
Poor*Clean Environment	0.020	0.07
Fair*Clean Environment	0.00	0.74
Good*Clean Environment	−0.01	0.16
Very Good*Clean Environment	−0.01	0.30
Poor*Transportation	0.01	0.61
Fair*Transportation	0.13	0.22
Good*Transportation	0.00	0.98
Very Good*Transportation	0.01	0.46
Poor*Social Environment	0.02	< 0.01
Fair*Social Environment	0.01	0.29
Good*Social Environment	−0.01	0.46
Very Good*Social Environment	0.00	0.78

^a^ HPI=Health Places Index. ^b^ Hierarchical linear modelling controlling for gender, age, race/ethnicity, HPI domain, and self-rated health Note: Omitted reference category: excellent self-rated health.

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
