# Peer review of "Community Advantage and Individual Self-Efficacy Promote Disaster Preparedness: A Multilevel Model among Persons with Disabilities"

_ijerph, 2019, doi:10.3390/ijerph16152779_

Round 1

Reviewer 1 Report

The manuscript is written based on vast amount of data and illustrates meticulous statistical analysis in proving the evidences to support the assumptions and hypotheses of what association factors might drive and motivate people with disabilities in preparing for disasters. In the current state of the manuscript, the type of the disasters are mentioned whether it is for natural/man-made or human-induced (in general); although I believe that a certain type of disaster may require a certain and specific type of approach, a specific type of coping mechanism, including disaster preparedness, especially for people with disabilities (whether people are limited in health and/or functional issues). For example, for people that are impaired in seeing, the disaster preparedness of printed communications via pamphlets, posters, etc. need to be tailored.

However, it is of great pleasure to read and being able (after some time) to understand the manuscript. No major comments are needed to add to this manuscript. But allow me to raise few complements. In Abstract part, line 27, following the phrase …”Emergent Threats Survey and the we”. It seems that there is a word missing between “the” and “we”.  Followingly, in the conclusions part, it is worth to illustrate (adding the graph/figure) how the results of the analysis will mirror back to the conceptual model linking disability and disaster preparedness.

Author Response

The manuscript is written based on vast amount of data and illustrates meticulous statistical analysis in proving the evidences to support the assumptions and hypotheses of what association factors might drive and motivate people with disabilities in preparing for disasters. In the current state of the manuscript, the type of the disasters are mentioned whether it is for natural/man-made or human-induced (in general); although I believe that a certain type of disaster may require a certain and specific type of approach, a specific type of coping mechanism, including disaster preparedness, especially for people with disabilities (whether people are limited in health and/or functional issues). For example, for people that are impaired in seeing, the disaster preparedness of printed communications via pamphlets, posters, etc. need to be tailored.

Response: Thank you for your review. We appreciate your comment regarding how disaster type might impact coping mechanisms among people with disabilities. We have addressed this comment in the limitations section.

However, it is of great pleasure to read and being able (after some time) to understand the manuscript. No major comments are needed to add to this manuscript. But allow me to raise few complements. In Abstract part, line 27, following the phrase …”Emergent Threats Survey and the we”. It seems that there is a word missing between “the” and “we”. 

Response: Thank you for bringing this to our attention. We have revised the abstract to fix this error.

Followingly, in the conclusions part, it is worth to illustrate (adding the graph/figure) how the results of the analysis will mirror back to the conceptual model linking disability and disaster preparedness.

Response: We have incorporated your suggested by adding a new figure (figure 3) to the discussion section that summarizes our findings.

Reviewer 2 Report

This is an interesting research to focus on disaster preparedness using multilevel analysis method.

Special comment:

Table 4 should be reaaranged and it is hard to read in its current form. The standardized beta coefficients for “Poor” is less than minus one, which rarely happen in regression analysis. This is mostly likely due to high correlation among input variables. Please check this issue and address this problem using suitable statistical method. The problem exists in Table 3.

Author Response

This is an interesting research to focus on disaster preparedness using multilevel analysis method.

Response: Thank you for your review.

Table 4 should be rearranged and it is hard to read in its current form.

Response: Thank you for bringing this to our attention. We have rearranged the table to improve clarity.

 The standardized beta coefficients for “Poor” is less than minus one, which rarely happen in regression analysis. This is mostly likely due to high correlation among input variables. Please check this issue and address this problem using suitable statistical method. The problem exists in Table 3.

Response: Thank you for catching this issue. This is a labeling error as the coefficient estimates are not standardized. We are using the proc glimmix procedure with the quadrature method, which derives estimates using the marginal log likelihood with an adaptive Gauss-Hermite quadrature. While this procedure allows us to run with multilevel model with survey weights, it does not allow for a standardized coefficient option. We have revised the table heading. For more information on this method, please see https://support.sas.com/documentation/cdl/en/statugglmmix/61788/PDF/default/statugglmmix.pdf

Reviewer 3 Report

Overall, the manuscript seems to be good. There are just a few points to modify.

The number of digits after decimal point is too long. Please round statistic values off to one or two decimal places.

Standardized beta coefficients are described on Table 3 and 4. Why some coefficients are more than one in spite of "standardized" coefficients?

Author Response

Overall, the manuscript seems to be good. There are just a few points to modify.

Response: Thank you for your review.

The number of digits after decimal point is too long. Please round statistic values off to one or two decimal places.

Response: We have revised the tables so that all numbers are rounded to two decimal places.

Standardized beta coefficients are described on Table 3 and 4. Why some coefficients are more than one in spite of "standardized" coefficients?

Response: Thank you for catching this issue. This is a labeling error as the coefficient estimates are not standardized. We are using the proc glimmix procedure with the quadrature method, which derives estimates using the marginal log likelihood with an adaptive Gauss-Hermite quadrature. While this procedure allows us to run with multilevel model with survey weights, it does not allow for a standardized coefficient option. We have revised the table heading. For more information on this method, please see https://support.sas.com/documentation/cdl/en/statugglmmix/61788/PDF/default/statugglmmix.pdf